# Relationship between anaemia, coagulation parameters during pregnancy and postpartum haemorrhage at childbirth: a prospective cohort study

Manisha Nair  ,[1] Shakuntala Chhabra,[2] Saswati Sanyal Choudhury,[3] Dipika Deka,[4] Gitanjali Deka,[5] Swapna D Kakoty,[6] Pramod Kumar,[2] Pranabika Mahanta,[7] Robin Medhi,[6] Anjali Rani,[8] Seeresha Rao,[9] Indrani Roy,[10] Carolin Solomi V,[11] Ratna Kanta Talukdar,[7] Farzana Zahir,[12] Nimmi Kansal,[13] Anil Arora,[13] Charles Opondo ,[14] Jane Armitage,[14] Michael Laffan,[15] Simon Stanworth,[16] Maria Quigley,[17] Colin Baigent,[14] Marian Knight,[17] Jennifer J Kurinczuk,[17] on behalf of the MaatHRI collaborators

SC, SSC, DD, GD, SDK, PK, PM, RM, AR, SR, IR, CSV, RKT and FZ contributed equally.

For numbered affiliations see end of article.

**Correspondence to**
Dr Manisha Nair;
manisha.nair@npeu.ox.ac.uk

## ABSTRACT

**Objectives** To investigate the association between coagulation parameters and severity of anaemia (moderate anaemia: haemoglobin (Hb) 7–9.9 g/dL and severe anaemia: Hb <7 g/dL) during pregnancy and relate these to postpartum haemorrhage (PPH) at childbirth.

**Design** A prospective cohort study of pregnant women recruited in the third trimester and followed-up after childbirth.

**Setting** Ten hospitals across four states in India.

**Participants** 1342 pregnant women.

**Intervention** Not applicable.

**Methods** Hb and coagulation parameters: fibrinogen, D-dimer, D-dimer/fibrinogen ratio, platelets and international normalised ratio (INR) were measured at baseline. Participants were followed-up to measure blood loss within 2 hours after childbirth and PPH was defined based on blood loss and clinical assessment. Associations between coagulation parameters, Hb, anaemia and PPH were examined using multivariable logistic regression models.

**Outcomes measures** Adjusted OR with 95% CI.

**Results** In women with severe anaemia during the third trimester, the D-dimer was 27% higher, mean fibrinogen 117 mg/dL lower, D-dimer/fibrinogen ratio 69% higher and INR 12% higher compared with women with no/ mild anaemia. Mean platelets in severe anaemia was 37.8×10$^9$/L lower compared with women with moderate anaemia. Similar relationships with smaller effect sizes were identified for women with moderate anaemia compared with women with no/mild anaemia. Low Hb and high INR at third trimester of pregnancy independently increased the odds of PPH at childbirth, but the other coagulation parameters were not found to be significantly associated with PPH.

**Conclusion** Altered blood coagulation profile in pregnant women with severe anaemia could be a risk factor for PPH and requires further evaluation.

## STRENGTHS AND LIMITATIONS OF THIS STUDY

⇒ This is the first study to investigate the role of coagulation in relation to the increased risk of postpartum haemorrhage (PPH) in women with moderate/severe anaemia.

⇒ The large prospective cohort study substantially removed the potential for reverse causation when estimating the effect of the coagulation parameters on PPH.

⇒ Another strength is reproducibility because we examined the relationship of haemoglobin (Hb) with five different parameters of coagulation and all suggested a similar effect.

⇒ The follow-up rate was 88% and the mean Hb concentration at baseline (exposure of interest) for 12% participants who could not be followed-up was not different from the participants who were followed-up.

## INTRODUCTION

Moderate (haemoglobin (Hb) 7–9.9 g/dL)[1] and severe anaemia (Hb <7 g/dL) during pregnancy not only increase the risk of postpartum haemorrhage (PPH), but also increase the risk of dying from PPH by several fold.[2–4] The public health problem of anaemia during pregnancy is graded as moderate–severe in 183 countries across the world with about 32.4 million (95% CI: 28.4 to 36.2) pregnant women with anaemia in 2011, globally.[5] Explanations include low Hb associated with reduced oxygen availability resulting in reduced uterine contractility and early fatigue causing uterine atony and PPH.[6] However, changes in the coagulation profile in anaemic pregnant women may also predispose them to an increased risk of bleeding.

Pregnancy is a state of physiological hypercoagulability, with an increase in fibrinogen and decrease in fibrinolytic activity, with increasing gestational age.[7 8] An expansion in plasma volume results in a physiological decrease in platelets, haematocrit and Hb during pregnancy,[7] although the prothrombin time remains largely stable.[7 9 10] While the haemostatic changes in normal pregnancy are well described, there have been few investigations of the relationship and potential clinical implications of coagulation abnormalities in association with severe anaemia in pregnant women. D-dimer was shown to be useful in risk stratification to rule out pulmonary embolism and to limit exposure of suspected pregnant women to imaging,[11] but the role of D-dimer and other coagulation parameters in risk stratification of PPH is not clear.

The primary objective of this study was to investigate the association between blood coagulation parameters (fibrinogen, D-dimer, D-dimer/fibrinogen ratio, platelets and international normalised ratio (INR)) and severity of anaemia during the third trimester of pregnancy. The secondary objective was to examine the relationship between anaemia and coagulation parameters during the third trimester and PPH at childbirth.

## METHODS
### Study design
A hospital-based prospective cohort study undertaken through the Maternal and perinatal Health Research collaboration, India (MaatHRI).[12]

### Study population
All pregnant women >28 weeks of gestation, ≥18 years of age and planning a vaginal birth in 10 MaatHRI collaborating hospitals across 4 states in India (Assam, Meghalaya, Uttar Pradesh and Maharashtra) were approached to participate in the study. The response rate was 99.8% and 1342 eligible pregnant women who provided written informed consent were recruited between October 2018 and May 2019. The women were followed-up during labour and childbirth and up to 48 hours post partum.

### Baseline data
Information was collected from women during the baseline assessment about sociodemographic characteristics, previous and current pregnancy problems, medical comorbidities and other pregnancy characteristics. Blood samples were collected to measure Hb, haematocrit, fibrinogen, D-dimer, platelets, prothrombin time (from which INR was derived) and if present, cause of anaemia (inferred from measurement of serum ferritin and Hb electrophoresis). Using the WHO definition for anaemia in pregnancy,[1] women with Hb ≥10 g/dL were classified as no/mild anaemia, 7–9.9 g/dL as moderate and <7 g/dL as severe anaemia. We generated a D-dimer/fibrinogen ratio that was used in other studies.[8 13 14] Since INR is not influenced by the pregnancy state,[9 10] using the standard cut-off, we classified pregnant women into high (>1.1)

and low INR (≤1.1) groups. Other coagulation parameters were analysed as continuous variables.

### Laboratory methods
The MaatHRI platform has a laboratory infrastructure through a partnership with a private laboratory in India.[12] Blood collection, processing, storage and analysis were standardised. All blood samples were analysed at the national laboratory. The assay methods, traceability and performance characteristics for each test were agreed with experts from the University of Oxford's Wolfson laboratory and the Indian laboratory partner. Online supplemental table S1 shows the traceability and in online supplemental table S2 we present the assay methods and their performance characteristics.

The laboratory measured time in transit for each sample and their quality. Depending on the remoteness of the hospital, the transit time ranged between 12 and 72 hours. Three types of samples were collected: EDTA whole blood for Hb, haematocrit, platelets and Hb electrophoresis, serum for ferritin and citrated plasma for D-dimer, fibrinogen and INR. Citrated plasma samples were centrifuged at 3700 revolutions per minute for 10 min using centrifuge machines of same make and model in all study hospitals, and aliquot was prepared from supernatant plasma, frozen immediately and shipped with dry ice. Samples that were inadequate, in terms of quantity, stability, temperature and other quality indicators, were discarded. A variable, 'hospital-code', was generated to account for transit time and other known and unknown potential biases related to sample quality in the statistical analysis.

### Follow-up data
Similar to other studies,[15 16] a calibrated blood collection drape was used to objectively measure the amount of blood loss within 2 hours after childbirth. The same make and model of drape (PPH alert bag) was used in all study hospitals. The drape was placed immediately after the birth of the baby (before removing the placenta) and blood loss was measured from the calibrated and colour coded markings on the drape. While maximum blood loss is just before and after the removal of placenta and up to 1 hour after childbirth, the drape was left in situ up to 2 hours during the post-birth observation period in the labour room if the woman continued to bleed.[16] The drape could not be used for women who had a caesarean section, in which case estimates of blood loss were measured by the obstetrician from the suction bottle and soaked sponges. Only pregnant women with a planned vaginal birth were recruited in the study, thus the participants who had a caesarean section were women who had an emergency section after spontaneous rupture of membranes. As a result, the suction bottle contained very little liquor thereby making the blood loss estimates more accurate. The objective measurement methods were in line with the recommendations of

the American College of Obstetricians and Gynecologists (ACOG).[17] ACOG acknowledges the difficulty in accurately measuring blood loss after childbirth, but recommends use of calibrated drapes and hospital-based protocols for collecting and measuring blood loss after childbirth, which are more accurate than visual estimation.[17]

PPH was defined based on measured blood loss within 2 hours after childbirth (≥500 mL for women who had a vaginal birth and ≥1000 mL for women who had a caesarean birth) and clinician diagnosed PPH requiring management. This was similar to the methods used to define PPH in other studies.[15 16] We also collected information about the mode of birth, maternal complications at birth, admission to intensive care unit and maternal death.

## Sample size

A priori sample size calculations were done for two primary parameters: D-dimer and fibrinogen (see online supplemental table S3). Sample sizes were calculated for a range of expected changes in the mean concentrations of the parameters (10%, 20% and 30%) between the no/mild anaemia and moderate/severe anaemia groups taking power (1-β)=90%, α=5% (two-tailed) and n1=n2. A sample size of 1028 had adequate power to detect a mean difference of 10% in the concentration of D-dimer and fibrinogen between the two study groups assuming a mean of 0.11 mg/dL (SD=0.573) for D-dimer[18] and 379 mg/dL (SD=0.78) for fibrinogen[19] in the baseline groups. This was inflated by 15% to account for potential loses which led to a total sample of 1209, rounded off to 1200 (n1=n2=600). However, we were able to increase the sample size during the recruitment phase, finally recruiting 1342 pregnant women, which allowed us to examine the difference in the concentration of coagulation parameters between three groups: no/mild anaemia, moderate and severe anaemia.

## Statistical analysis

Descriptive statistics were used for all blood parameters, participant characteristics at baseline and PPH at childbirth. We calculated and compared the mean Hb across categories of gestational age, and mean gestational age across the categories of anaemia using t-test with Bartlett's statistics for equal variances. We examined the distribution of the continuous variables, and blood parameters that were not normally distributed: D-dimer, D-dimer/fibrinogen ratio and INR, were log transformed to create a normal distribution. These were used as outcome variables in the primary analysis and multivariable linear regression models were used to examine their individual association with Hb and anaemia after controlling for known confounding variables including gestational age, maternal age, hypertensive disorders of pregnancy (which included gestational hypertension, pre-eclampsia, eclampsia, superimposed pre-eclampsia on chronic hypertension as well as severe forms of pre-eclampsia such as

haemolysis, elevated liver enzymes and low platelet count (HELLP) syndrome), pre-existing medical problems and hospital-code. We conducted tests for linear trend and used $\chi^2$ tests to assess heterogeneity in ORs across categories of anaemia. We also examined the presence of any non-linear relationships between Hb and the coagulation parameters. The analysis was repeated using haematocrit instead of Hb as the exposure variable to test reproducibility of the results.

For the secondary objective, we analysed the association of PPH with Hb, anaemia and the coagulation parameters using multivariable logistic regression analysis controlling for potential confounders and exploring significant interactions. We found the variable 'hospital-code' to be strongly correlated with hypertensive disorders of pregnancy and PPH. To improve model parsimony, 'hospital-code' was not included in the multivariable analysis. To understand whether the effect of Hb on PPH was moderated or mediated by each of the coagulation parameters, we tested for interaction and conducted mediation analysis, respectively. Likelihood-ratio test was used to examine statistically significant interactions at p<0.1 considering a lower power for the subgroup analysis. Mediation analysis was undertaken using the generalisation to the Baron-Kenny approach to mediation analysis.[20]

Missing data for the blood parameters were related to samples being discarded due to quality issues, but not with the level of the parameter itself or Hb. Therefore, data in the study were considered missing at random and complete case analysis was used. All results were considered significant at a two-tailed p value of <0.05. Analyses were undertaken using Stata V.16, special edition (StataCorp).

## Patient and public involvement

Patient and public were not involved in the design, conduct or reporting of the study.

## Participant consent

Written informed consent was taken from all participants.

## RESULTS

The baseline characteristics of the study population are presented in table 1. The mean Hb was 10.3 g/dL and mean gestational age at recruitment was 35.3 weeks. Mean Hb was similar in the different periods of gestational age (p=0.275, see table 1) and gestational age was also similar across the categories of anaemia (p=0.128, see table 1). The proportions of no/mild, moderate and severe anaemia in the study population were 58.9%, 34.7% and 5.3%, respectively. The most common anaemia was iron deficiency (microcytic–hypochromic anaemia 19.8% and about 28.5% had serum ferritin <15 µg/L), but 12.5% of women had macrocytic anaemia and 14% had an HbE (Haemoglobin E) trait or disease. The mean haematocrit was 21% in women with severe anaemia compared with

**Table 1** Baseline characteristics of the study population

| Characteristics at baseline (total participants at baseline) | Overall study population | No/mild anaemia (Hb ≥10 g/dL) N=790 | Moderate anaemia (Hb 7–9.9 g/dL) N=465 | Severe anaemia (Hb <7 g/dL) N=71 | Missing Hb information N=16 |
|---|---|---|---|---|---|
| | **Mean (SD)** | | | | |
| Maternal age (in years); N=1334 | 24.5 (4.2) | 24.5 (4.0) | 24.6 (4.4) | 24.9 (4.5) | 24.2 (4.5) |
| Gestational age at baseline recruitment (in weeks); N=1342 | 35.3 (3.7) | 35.4 (3.7) | 35.1 (3.8) | 35.0 (3.7) | 34.9 (3.2) |
| **Blood parameters at baseline (unit of measure)** | **Mean (SD)** | | | | |
| Hb (Hb in g/dL); N=1326 | 10.3 (1.9) | – | – | – | – |
| Hb in g/dL by categories of gestational age (p=0.275) | | | | | – |
| 28–32 weeks | 10.0 (1.8) | – | – | – | – |
| 33–36 weeks | 10.5 (1.9) | – | – | – | – |
| ≥37 weeks | 10.4 (1.9) | – | – | – | – |
| Platelets (×$10^9$ /L); N=1305 | 195.6 (73.7) | 187.4 (2.5) | 213.5 (3.6) | 168.3 (9.4) | – |
| Fibrinogen (mg/dL); N=1270 | 410.9 (129.2) | 431.1 (4.8) | 390.5 (5.7) | 319.9 (12.7) | 339.1 (98.6) |
| | **Median (IQR)** | | | | |
| D-dimer (mg/dL); N=1264 | 0.08 (0.07) | 0.07 (0.06) | 0.08 (0.07) | 0.10 (0.09) | 0.12 (0.13) |
| D-dimer/fibrinogen ratio; N=1250 | 0.0002 (.0002) | 0.0002 (0.0002) | 0.0002 (0.0002) | 0.0003 (0.0005) | 0.0005 (0.0005) |
| International normalised ratio (INR); N=1243 | 0.96 (0.12) | 0.94 (0.13) | 0.96 (0.1) | 0.99 (0.13) | 0.96 (0.31) |
| Body mass index at first antenatal check-up (kg/m²); N=1165 | 21.1 (4.3) | 21.2 (4.6) | 20.8 (4.4) | 21.1 (4.2) | 21.2 (3.8) |
| | **No. of women (%)** | | | | |
| INR | | | | | |
| ≤1.1 | 1126 (83.9) | 679 (86.0) | 394 (84.7) | 49 (69.0) | 4 (25.0) |
| >1.1 | 117 (8.7) | 61 (7.7) | 40 (8.6) | 14 (19.7) | 2 (12.5) |
| Missing | 99 (7.4) | 50 (6.3) | 31 (6.7) | 8 (11.3) | 10 (62.5) |
| HbE | | | | | |
| Normal | 1141 (85.0) | 689 (87.2) | 390 (83.9) | 62 (87.3) | 0 (0) |
| Trait | 129 (9.6) | 84 (10.6) | 38 (8.2) | 7 (9.9) | 0 (0) |
| Disease | 56 (4.2) | 17 (2.2) | 37 (7.9) | 2 (2.8) | 0 (0) |
| Missing | 16 (1.2) | 0 (0) | 0 (0) | 0 (0) | 16 (100.0) |
| Microcytic–hypochromic anaemia | | | | | |
| No | 1010 (75.3) | 684 (86.6) | 293 (63.0) | 33 (46.5) | 0 (0) |
| Yes | 266 (19.8) | 84 (10.6) | 155 (33.3) | 27 (38.0) | 0 (0) |
| Missing | 66 (4.9) | 22 (2.8) | 17 (3.7) | 11 (15.5) | 16 (100.0) |
| Macrocytic anaemia | | | | | |
| No | 1110 (82.7) | 647 (81.9) | 413 (88.8) | 50 (70.4) | 0 (0) |
| Yes | 168 (12.5) | 121 (15.3) | 36 (7.7) | 11 (15.5) | 0 (0) |
| Missing | 64 (4.8) | 22 (2.8) | 16 (3.4) | 10 (14.1) | 16 (100.0) |
| Serum ferritin | | | | | |
| ≥15 µg/L | 927 (69.1) | 612 (77.5) | 269 (57.8) | 39 (54.9) | 7 (43.7) |
| <15 µg/L | 383 (28.5) | 165 (20.9) | 190 (40.9) | 28 (39.4) | 0 (0) |

Continued

**Table 1** Continued

| Characteristics at baseline (total participants at baseline) | Overall study population | No/mild anaemia (Hb ≥10 g/dL) N=790 | Moderate anaemia (Hb 7–9.9 g/dL) N=465 | Severe anaemia (Hb <7 g/dL) N=71 | Missing Hb information N=16 |
|---|---|---|---|---|---|
| Missing | 32 (2.4) | 13 (1.6) | 6 (1.3) | 4 (5.6) | 9 (56.3) |
| **Other pregnancy characteristics at baseline** | | | | | |
| Hypertensive disorders of pregnancy | | | | | |
| No | 1290 (96.1) | 761 (96.3) | 447 (96.1) | 66 (93.0) | 16 (100) |
| Yes | 46 (3.4) | 24 (3.0) | 17 (3.7) | 5 (7.0) | 0 (0) |
| Missing | 6 (0.5) | 5 (0.6) | 1 (0.2) | 0 (0) | 0 (0) |
| Pre-existing medical problems (other than haemoglobinopathies) | | | | | |
| No | 1281 (95.5) | 759 (96.1) | 443 (95.3) | 63 (88.7) | 16 (100) |
| Yes | 58 (4.3) | 29 (3.7) | 21 (4.5) | 8 (11.3) | 0 (0) |
| Missing | 3 (0.2) | 2 (0.2) | 1 (0.2) | 0 (0) | 0 (0) |

Pre-existing medical problems (excluding haemoglobinopathies) included diabetes, essential hypertension, rheumatic heart disease, hypothyroidism, urinary tract infection, kidney stone, appendicitis, gall bladder problems, ovarian tumour, pulmonary tuberculosis and hepatitis C infection.
Hb, haemoglobin ; HbE, Haemoglobin E.

30% in women with moderate anaemia and 37% in women with mild/no anaemia. About 17% of the study population reported a problem during the current pregnancy. A total of eight women reported an antepartum haemorrhage, of these four were in the category of no/mild anaemia, three in moderate and one in the severe anaemia group.

Key follow-up data are presented in table 2. There was a 12% loss to follow-up, but no difference in mean Hb during the third trimester between women who were followed-up (10 g/dL) and those not followed-up (10 g/dL). A flow chart showing the study population is provided in online supplemental figure S1.

### Association of coagulation parameters with Hb and anaemia

The results of the linear regression analyses are presented in table 3 and figures 1–6. All coagulation parameters were significantly associated with Hb and anaemia during the third trimester. The relationships were linear (inverse linear associations), except for platelets that had a non-linear inverted J-shaped association with Hb (figure 1).

After adjustment, the D-dimer concentration was 8% (95% CI –1% to +17%) higher in women with moderate anaemia and 27% (95% CI 7% to 50%) higher in severe anaemia compared with no/mild anaemia (p value for linear trend=0.003). In women with moderate anaemia, the mean fibrinogen concentration was 39.2 mg/dL (95% CI 24.9 to 53.7 mg/dL) lower and in severe anaemia 117.2 mg/dL (95% CI 86.1 to 148.3 mg/dL) lower than in women with no/mild anaemia (p value for trend <0.001). Consequently, the D-dimer/fibrinogen ratio was 17% (95% CI 6% to 29%) and 69% (95% CI 36% to 100%) higher, respectively, in women with moderate and severe anaemia compared with women with no/mild anaemia (p value for trend <0.001).

Given the inverted J-shaped association between Hb and platelets, the moderate anaemia group was

| Follow-up (total participants at follow-up=1178, 12% loss to follow-up) | Overall study population followed-up N=1178 | No/mild anaemia (Hb ≥10 g/dL) N=709 | Moderate anaemia (Hb 7–9.9 g/dL) N=390 | Severe anaemia (Hb <7 g/dL) N=66 | Missing Hb information N=13 |
|---|---|---|---|---|---|
| **Table 2** Key data from the follow-up | | | | | |
| PPH | No. of women (%) | | | | |
| No | 1159 (98.4) | 701 (98.9) | 382 (97.9) | 63 (95.5) | 13 (100) |
| Yes | 19 (1.6) | 8 (1.1) | 8 (2.1) | 3 (4.5) | 0 (0) |
| Mode of delivery | | | | | |
| Vaginal birth | 852 (72.3) | 515 (72.6) | 279 (71.4) | 49 (74.2) | 9 (69.2) |
| Caesarean birth | 326 (27.7) | 194 (27.4) | 111 (28.5) | 17 (25.8) | 4 (30.8) |

Hb, haemoglobin; PPH, postpartum haemorrhage.

**Table 3** Association of coagulation parameters with Hb and anaemia at third trimester

| Independent variables | Outcome variables | | |
|---|---|---|---|
| | Unadjusted coefficient (95% CI) | Adjusted* coefficient (95% CI) | P value—test for linear trend |
| **D-dimer (mg/dl)** | | | |
| Hb | 0.96 (0.94 to 0.97) | 0.96 (0.94 to 0.98) | <0.001 |
| No/mild anaemia | 1 (ref)† | 1 (ref)† | 0.003‡ |
| Moderate anaemia | 1.11 (1.02 to 1.20) | 1.08 (0.99 to 1.17) | |
| Severe anaemia | 1.25 (1.04 to 1.50) | 1.27 (1.07 to 1.50) | |
| **Fibrinogen (mg/dl)** | | | |
| Hb | 14.68 (11.11 to 18.24) | 15.58 (12.08 to 19.09) | <0.001 |
| No/mild anaemia | 0 (Ref) | 0 (Ref) | <0.001‡ |
| Moderate anaemia | −40.5 (−55.3 to −25.7) | −39.2 (−53.7 to −24.9) | |
| Severe anaemia | −111.1 (−143.4 to −78.9) | −117.2 (−148.3 to −86.1) | |
| **D-dimer/fibrinogen ratio** | | | |
| Hb | 0.92 (0.90 to 0.95) | 0.93 (0.91 to 0.95) | <0.001 |
| No/mild anaemia | 1 (ref)† | 1 (ref)† | <0.001‡ |
| Moderate anaemia | 1.21 (1.09 to 1.34) | 1.17 (1.06 to 1.29) | |
| Severe anaemia | 1.63 (1.30 to 2.06) | 1.69 (1.36 to 2.09) | |
| **International normalised ratio** | | | |
| Hb | 0.98 (0.97 to 0.99) | 0.99 (0.98 to 0.99) | 0.001 |
| No/mild anaemia | 1 (ref)† | 1 (ref)† | 0.007‡ |
| Moderate anaemia | 1.03 (0.99 to 1.06) | 1.02 (0.98 to 1.05) | |
| Severe anaemia | 1.09 (1.02 to 1.17) | 1.12 (1.04 to 1.19) | |
| **Platelets (X $10^9$/L)** | | | |
| Hb | −3.79 (−5.83 to −1.74) | −4.57 (−6.64 to −2.49) | NA |
| No/mild anaemia | −26.04 (−34.40 to −17.68) | −26.09 (−34.51 to −17.67) | NA |
| Moderate anaemia | 0 (ref) | 0 (ref) | |
| Severe anaemia | −45.21 (−63.91 to −26.51) | −37.78 (−56.47 to −19.09) | |

*Adjusted for gestational age, maternal age, hypertensive disorders of pregnancy, pre-existing medical problems and hospital-code.
†Exponent of Log, hence reference is '1' (instead of 0)
‡P value—test for linear trend.
Hb, haemoglobin; NA, not applicable.

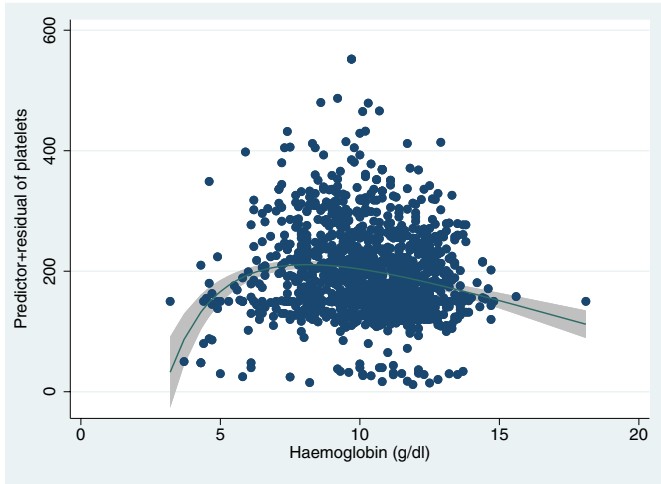

**Figure 1** Inverted J-shaped association between Hb and platelets. Hb, haemoglobin.

taken as the comparator. Compared with women with moderate anaemia, those with no/mild anaemia had a mean platelet concentration 26×$10^9$/L (95% CI 17.7 to 34.5 ×$10^9$/L) lower and those with severe anaemia 38×$10^9$ /L (19.1 to 56.5 ×$10^9$ /L) lower. The INR was 2% (95% CI −2% to 5%) and 12% (95% CI 4% to 19%) higher in women with moderate and severe anaemia, respectively, compared with women with no/mild anaemia. The odds of having a high INR (>1.1) decreased by 19% per g/dL increase in Hb (adjusted OR (aOR) 0.81, 95% CI 0.73 to 0.91, p<0.001). The odds of having a high INR in women with moderate anaemia was not significantly different from women with no/mild anaemia (aOR 1.12, 95% CI 0.69 to 1.84, p=0.647), but women with severe anaemia had more than fivefold higher odds of having a high INR (aOR 5.10, 95% CI 2.31 to 11.29, p<0.001).

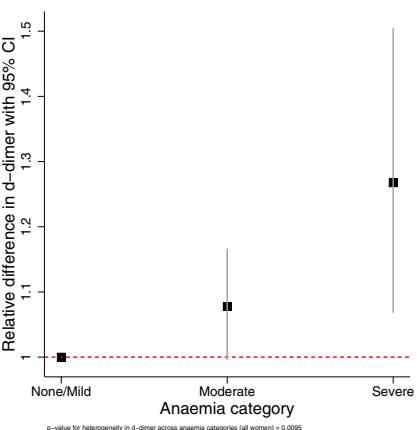

**Figure 2** Relative difference in D-dimer across the categories of anaemia.

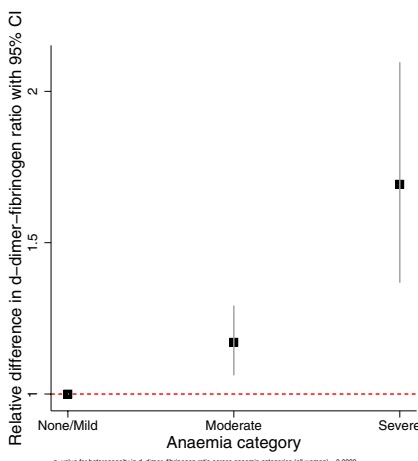

**Figure 4** Relative difference in D-dimer/fibrinogen ratio across the categories of anaemia.

The tests for heterogeneity showed that all ORs were significantly different across the categories of anaemia. Figures 2–6 show the relationship between the coagulation parameters and the categories of anaemia. The findings did not change when stratified by types of anaemia, although the 95% CI widened due to the small numbers in each stratified category. Furthermore, repeating the analyses using haematocrit as the exposure variable did not change the results materially (online supplemental table S4).

### Association of PPH at childbirth with Hb and anaemia in the third trimester of pregnancy

After adjusting for known confounders, the odds of having a PPH at childbirth *increased* by 22% per g/dL *decrease* in Hb (aOR 0.78, 95% CI 0.63 to 0.98). The adjusted odds of having a PPH was nearly twofold higher in women with moderate anaemia and more than fivefold higher in women with severe anaemia compared with women with mild/no anaemia. There was a significant linear trend of increasing adjusted

odds of PPH with increasing severity of anaemia (p value for linear trend 0.035) (table 4).

### Association of PPH at childbirth with coagulation parameters in the third trimester of pregnancy

After adjusting for confounders, the odds of having a PPH increased by more than fivefold in women who had an INR >1.1 during the third trimester of pregnancy (table 5). The other coagulation parameters, D-dimer, fibrinogen and platelets were not significantly associated with PPH at childbirth (table 5). Mediation analysis showed no significant mediation of the effect of Hb on PPH via any coagulation parameter. There was a pattern of increasing predicted probability of PPH with a decrease in Hb and increase in D-dimer (online supplemental figure S2). Nevertheless we did not find evidence of statistical interaction between Hb and D-dimer in their association with PPH (p value 0.529). We did not find any significant interaction between Hb and the other coagulation parameters.

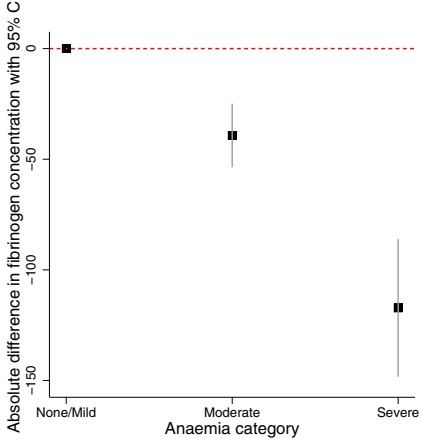

**Figure 3** Absolute difference in fibrinogen across the categories of anaemia.

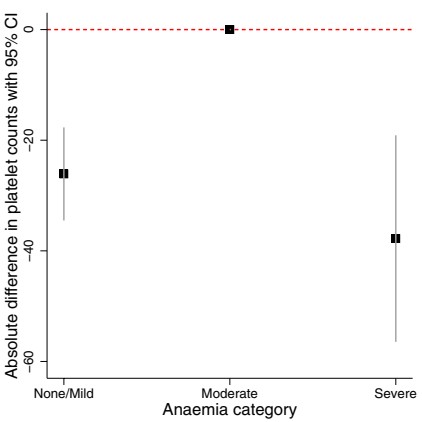

**Figure 5** Absolute difference in platelets across the categories of anaemia.

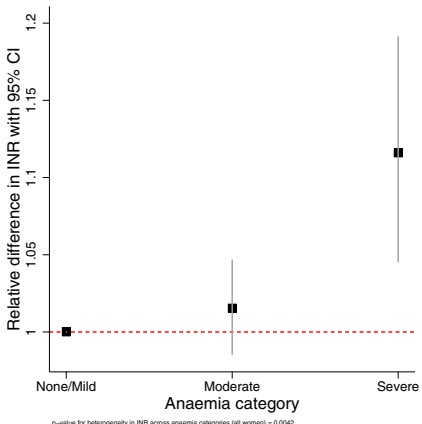

**Figure 6** Relative difference in INR across the categories of anaemia. INR, international normalised ratio.

## DISCUSSION

The study showed that pregnant women with severe anaemia during the third trimester of pregnancy had a higher D-dimer, lower fibrinogen and therefore a higher D-dimer/fibrinogen ratio than those with mild or moderate anaemia as well as a higher INR, after controlling for known confounders. Similar associations were observed among women with moderate anaemia with levels intermediate between severe and mild anaemia. Having a lower Hb and high INR (>1.1) during the third trimester of pregnancy was independently associated with higher odds of PPH, but we did not find any association between PPH and the other coagulation parameters.

Studies,[2 21] including our previous study in India,[3] have repeatedly shown that pregnant women with anaemia (particularly severe anaemia) are at a higher risk of PPH. It is also known that low fibrinogen, high INR and high D-dimer or other measures of fibrinolysis are associated with an increased risk of PPH[7] but to what extent these changes are associated with anaemia has not previously been described. The association between platelets and PPH is not clear,[7] nor whether it is the total concentration or functionality of platelets that matter.[7]

This study identified a new potential role of an impaired coagulation profile in pregnant women with anaemia that could lead to PPH. These potential associations are hypothesis generating for further research, both to

understand the direct causal effects and the mechanisms by which the coagulation changes might exert an impact on anaemic women at childbirth. The primary observation was lower fibrinogen level in women with moderate and severe anaemia in the study population. It is known that fibrinogen levels increase by more than 200% during pregnancy[7] compared with the non-pregnant state to prevent haemorrhage during childbirth, and a recent meta-analysis of concentration of coagulation parameters by gestational age in pregnancy estimated a mean fibrinogen level of 556 (466–664) mg/dL during the third trimester of pregnancy.[22] Compared with this, mean fibrinogen levels during the third trimester were 391 (379–402) mg/dL and 320 (295–345) mg/dL in pregnant women with moderate and severe anaemia, respectively, in the study population with a linear decrease in fibrinogen level by severity of anaemia, thus potentially increasing the risk of PPH.

**Table 5** Association between PPH at childbirth and coagulation parameters in the third trimester of pregnancy

| Outcome—PPH at childbirth | Predictors—coagulation parameters in the third trimester of pregnancy | |
| --- | --- | --- |
| | Unadjusted OR (95% CI) | Adjusted* OR (95% CI) |
| **D-dimer** | | |
| No | 1 (ref) | 1 (ref) |
| Yes | 1.00 (0.79 to 1.26) | 1.03 (0.80 to 1.32) |
| **Fibrinogen** | | |
| No | 1 (ref) | 1 (ref) |
| Yes | 1.00 (0.99 to 1.006) | 1.00 (0.99 to 1.005) |
| **Platelets** | | |
| No | 1 (ref) | 1 (ref) |
| Yes | 0.99 (0.98 to 1.001) | 0.99 (0.98 to 1.002) |
| **International normalised ratio >1.1** | | |
| No | 1 (ref) | 1 (ref) |
| Yes | 1.76 (0.39 to 7.78) | 5.74 (1.09 to 30.19) |

*Regression models adjusted for gestational age, maternal age, hypertensive disorders of pregnancy, pre-existing medical problems and mode of delivery; D-dimer in mg/L FEU; fibrinogen in mg/dL; platelets ×10$^9$/L.
PPH, postpartum haemorrhage .

**Table 4** Association of PPH at childbirth with haemoglobin and anaemia

| Independent variables | Outcome: PPH at childbirth | | |
| --- | --- | --- | --- |
| | Unadjusted OR (95% CI) | Adjusted* OR (95% CI) | P value—test for linear trend |
| Anaemia | | | |
| No/mild | 1 (ref) | 1 (ref) | 0.035 |
| Moderate | 1.84 (0.68 to 4.93) | 1.82 (0.66 to 5.01) | |
| Severe | 4.17 (1.08 to 16.12) | 5.11 (1.19 to 21.93) | |

*Adjusted for gestational age, maternal age, hypertensive disorders of pregnancy, pre-existing medical problems and mode of birth.
PPH, postpartum haemorrhage .

There is some evidence that haemodilution has a profibrinolytic effect,[23 24] thus another possibility is the presence of low grade pre-delivery fibrinolysis in pregnant women with severe anaemia in the study, which might also predispose them to higher blood loss or haemorrhage at childbirth. The median D-dimer levels in the study population in different categories of anaemia (table 3) was comparable with the estimated mean D-Dimer during the third trimester of pregnancy in the meta-analysis,[22] but we found a linear increase in D-dimer-to-fibrinogen ratio with increase in severity of anaemia. Under a conventional state of hypercoagulability during pregnancy, the decrease in fibrinogen should have been matched with a decrease in fibrinolytic activity, but in our study population with moderate and severe anaemia, the two processes seem to be operating in opposite directions, thereby creating a potential imbalance in clot formation and lysis which could increase the risk of PPH. Further, we also observed a pattern of low Hb and high D-dimer having a multiplicative effect on increased probability of PPH, although the interaction was not statistically significant. We did not find any underlying cause of blood loss (example placenta praevia or abruption), or antepartum haemorrhage in pregnant women with anaemia that could explain both low Hb and high D-dimer.

Likewise, the relative increase in INR in pregnant women with severe anaemia cannot be explained by the physiological changes in pregnancy as INR generally remains stable in pregnancy.[10] Women with severe anaemia had a low haematocrit (21%). While high haematocrit (>50%) is thought to artificially prolong PT (prothrombin time) from which INR is calculated, a low haematocrit (<25%) should not affect the measurement of PT using standard sodium citrate tubes.[25] It is possible that women with severe anaemia, who were mostly iron deficient, also have vitamin K deficiency due to malnutrition leading to an increase in INR. Prolongation of PT and increase in INR have been shown in patients with sickle cell disease, the increase being proportional to the severity of anaemia,[26] and in a study of patients with haematological malignancies who were treated with chemotherapy,[27] suggesting a delay in the initiation of the coagulation cascade in people with low Hb. This could explain the observed higher odds of PPH associated with high INR >1.1 in our study population.

We found an inverted J-shaped association between platelets and severity of anaemia. While the lower mean concentration of platelets in women with severe anaemia is in line with the impairment in the other coagulation parameters, the reasons for the lower mean concentration in no/mild anaemia compared with moderate anaemia is unclear. One possible explanation could be residual confounding by hypertensive disorders of pregnancy. Women who have a severe disorder (eg, HELLP syndrome) have low haemodilution (high Hb) and low platelets.[28] The relationship between anaemia and platelets is also unclear. In vitro studies show agglutination of platelets with lowering of Hb,[29 30] others found an

association between iron deficiency anaemia and thrombocytosis,[31 32] and yet others suggest that anaemia impairs the role of red blood cells that normally push the platelets towards the vessel wall during the coagulation process to initiate clot formation.[27 33]

## Strengths and limitations

The main strength of this study is that it was large and prospective allowing examination of the relationship between Hb, anaemia and coagulation parameters during late pregnancy and their subsequent effects on blood loss at childbirth. Robust and standardised methods were employed to minimise bias, and improve the validity and reliability of the findings. The design allowed adjustment for gestational age, a major factor influencing coagulation parameters. The blood parameters were measured prospectively in the same laboratory in the third trimester of pregnancy (baseline) prior to labour and birth, and blood loss was measured at childbirth, addressing the risk of reverse causality. Another strength is reproducibility. We examined the relationship of Hb with five different parameters of coagulation and all suggested the same effect. We were able to replicate the findings using haematocrit as the exposure variable.

The findings are generalisable to the population in India as data were collected from 10 hospitals across four states in India, which are different in terms of their socioeconomic contexts, healthcare facilities, food habits, prevalence of malnutrition and anaemia among pregnant women and burden of maternal complications and death. The physiological changes associated with anaemia observed in our study are likely to be generalisable to all pregnant women, globally.

One limitation was the 12% loss to follow-up due to staff problems in one hospital. None of the participants in that hospital were followed-up during or after childbirth, thus any bias due to loss to follow-up is likely to be minimal, as it was not related to the exposures or outcomes examined in the study. The mean Hb in women who were followed-up was not different from those who were not followed-up. Although we objectively measured blood loss at childbirth using a calibrated blood collection drape (for vaginal birth) and from suction bottle and soaked sponges (for caesarean birth), we cannot rule out measurement errors, but as mentioned earlier, the methods conformed to the recommendations of ACOG. In addition, there is no evidence that clinician estimated blood loss or blood loss measured by calibrated drape is associated with differential misclassification of PPH. Therefore, it is less likely that the results are influenced by the methods used for ascertaining PPH at childbirth. 1.6% of the study population had PPH which was comparable with the rate estimated in a previous study (1.1%),[34] but the lower number of events (n=19) reduced the statistical power of the analysis for the secondary objective, which we acknowledge as a limitation. Low number of events also limited the statistical power of the effect of the interaction between low Hb and high D-dimer

on increased probability of PPH. Further, despite using standardised laboratory procedures and accounting for time taken for the blood samples to reach the national reference laboratory from the study hospitals, we cannot completely rule out measurement errors for the blood parameters.

## CONCLUSION

In this study of pregnant women, measures of the coagulation parameters in the third trimester were significantly associated with the severity of anaemia. We identified a substantial independent effect of high INR and low Hb on increased risk of PPH at childbirth. Given the high prevalence of anaemia in pregnant women, globally, further studies are required to investigate the mechanisms through which coagulation parameters could increase the risk of PPH in pregnant women with anaemia.

**Author affiliations**
[1]NPEU, Nuffield Department of Population Health, Oxford University, Oxford, UK
[2]Department of Obstetrics and Gynaecology, Mahatma Gandhi Institute of Medical Sciences, Sevagram, Maharashtra, India
[3]Department of Obstetrics and Gynaecology, Gauhati Medical College and Hospital, Guwahati, Assam, India
[4]Srimanta Sankaradeva University of Health Sciences, Guwahati, Assam, India
[5]Department of Obstetrics and Gynaecology, Tezpur Medical College, Tezpur, India
[6]Department of Obstetrics and Gynaecology, Fakhruddin Ali Ahmed Medical College and Hospital, Barpeta, Assam, India
[7]Department of Obstetrics and Gynaecology, Jorhat Medical College and Hospital, Jorhat, Assam, India
[8]Department of Obstetrics and Gynaecology, Banaras Hindu University Institute of Medical Sciences, Varanasi, Uttar Pradesh, India
[9]Department of Obstetrics and Gynaecology, Silchar Medical College and Hospital, Silchar, Assam, India
[10]Department of Obstetrics and Gynaecology, Nazareth Hospital, Shillong, Meghalaya, India
[11]Department of Obstetrics and Gynaecology, Makunda Christian Leprosy and General Hospital, Karimganj, Assam, India
[12]Department of Obstetrics and Gynaecology, Assam Medical College, Dibrugarh, Assam, India
[13]National Reference Laboratory, Dr Lal Pathlabs, New Delhi, India
[14]Nuffield Department of Population Health, Oxford University, Oxford, UK
[15]Haemostasis and Thrombosis, Imperial College London Faculty of Medicine, London, UK
[16]Department of Haematology/Transfusion Medicine, Oxford University, Oxford, UK
[17]National Perinatal Epidemiology Unit, Oxford University, Oxford, UK

**Collaborators** Details about the MaatHRI collaboration can be found on https://www.npeu.ox.ac.uk/maathri.

**Contributors** MN developed the concept and designed the study, conducted the statistical analysis, led the overall work as chief investigator and wrote the first draft of the paper. SC, SSC, DD, GD, SDK, PK, PM, RM, AR, SR, IR, CSV, RKT and FZ contributed equally, and their names are included in the alphabetical order of their last name. They are collaborators and investigators for the study, contributed to developing the study and led the work in their respective institution. They also edited the paper. NK and AA contributed to developing the laboratory measures for the study, and AA edited the laboratory measurement section of the paper. CO provided statistical expertise, and contributed to writing the statistical methods and results. JA contributed to developing the concept for the study and edited the paper. SS and ML contributed to interpreting the results of the study and edited the paper. MQ provided statistical advice. CB, MK and JJK are advisors and have contributed to developing the study. JJK also edited the paper.

**Funding** The study was funded by a Nuffield Department of Population Health (NDPH) Pump-priming award, and the MaatHRI platform is funded by a Medical Research Council Career Development Award to MN (Grant Ref: MR/P022030/1). The funders had no role in the study design, data collection, analysis or writing of the report. MN had full access to all the information for the paper and had final responsibility for the decision to submit for publication.

**Competing interests** None declared.

**Patient consent for publication** Not applicable.

**Ethics approval** The study was approved by the institutional review boards of each coordinating Indian institution, namely: Srimanta Sankaradeva University of Health Sciences, Guwahati, Assam; Nazareth Hospital, Shillong, Meghalaya; Emmanuel Hospital Association, New Delhi; Mahatma Gandhi Institute of Medical Sciences, Sevagram, Maharashtra; and the Institute of Medical Sciences, Banaras Hindu University, Varanasi, Uttar Pradesh. It also received approval from the Government of India's Health Ministry's Screening Committee, the Indian Council of Medical Research, New Delhi and by the Oxford Tropical Research Ethics Committee (OxTREC), University of Oxford, UK.

**Provenance and peer review** Not commissioned; externally peer reviewed.

**Data availability statement** Data are available upon reasonable request. The data and metadata used in this study are available for free and can be obtained by contacting the corresponding author.

**ORCID iDs**
Manisha Nair http://orcid.org/0000-0003-0660-5054
Charles Opondo http://orcid.org/0000-0001-8155-4117

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
