## [Reviewer comments · BMJ Open]

ARTICLE DETAILS

TITLE (PROVISIONAL)	Relationship between anaemia, coagulation parameters during pregnancy and postpartum haemorrhage at childbirth: a prospective cohort study
AUTHORS	Nair, Manisha; Chhabra, Shakuntala; Choudhury, Saswati; Deka, Dipika; Deka, Gitanjali; Kakoty, Swapna; Kumar, Pramod; Mahanta, Pranabika; Medhi, Robin; Rani, Anjali; Rao, Seerasha; Roy, Indrani; Solomi V, Carolin; Talukdar, Ratna; Zahir, Farzana; Kansal, Nimmi; Arora, Anil; Opondo, Charles; Armitage, Jane; Laffan, Michael; Stanworth, Simon; Quigley, Maria; Baigent, Colin; Knight, Marian; Kurinczuk, Jennifer

VERSION 1 – REVIEW

REVIEWER	Joseph, K. S. Dalhousie Univ I have no competing interests to declare (except that I have previously published with one of the authors on the paper. Professor Marian Knight).
REVIEW RETURNED	16-May-2021

GENERAL COMMENTS	The authors carried out a prospective cohort study in 10 hospitals across India and investigated the association between coagulation indices and anaemia severity among pregnant women, and also between these characteristics and postpartum hemorrhage. 1342 pregnant women were included in the study and there was a 12% loss to follow up. The study showed associations between anaemia severity and D-dimer, fibrinogen, and INR and also postpartum hemorrhage. Comments 1. This study examined an important global health issue as anaemia in pregnancy is a significant problem and postpartum hemorrhage is common cause of maternal mortality and severe morbidity.2. In Table 1 and Table 2, it would be preferable to provide the characteristics of the study subjects and follow up information within the categories of interest viz., no/mild anaemia, moderate anaemia and severe anaemia. Such tabular analysis will allow readers to map means and frequencies to the unadjusted mean differences odds ratios from the regression analysis and better understand the results of the adjusted analysis.3. Of the 1178 subjects followed up and assessed for postpartum hemorrhage, 19 had this outcome. The results of the regression analysis in Table 4 and Table 5 (which involved adjustment for several variables) are limited by this relatively small number of
--

	outcomes and this should be acknowledged as a weakness. The authors could consider examining blood loss as opposed to postpartum hemorrhage as the outcome. Other comments: 4. Lines 7-9 “The public health problem of anaemia during pregnancy is graded as moderate-severe in 183 countries across the world with about 529 million pregnant women with anaemia in 2011, globally⁵ and an estimated 295,000 maternal deaths annually⁶.” It is not possible that there were 529 million pregnant women with anaemia in 2011 as currently there are only about 140 million births per year globally. That number refers to women of reproductive age with anaemia. This sentence also needs to be reworded as it implies that the 295,000 maternal deaths may be related to anaemia when in fact that is the total number of maternal deaths per year. 5. A clarification regarding the sample size calculation is warranted. Why did the authors assume equal sized groups with n1=n2? The primary objective was to compare across categories of anaemia (no/mild, moderate and severe). Also, when the authors refer to ‘change’ (Table S3), do they mean ‘difference’ between groups? 6. Column headings in Table S4 state that the numbers provided are odds ratios but several of them are negative numbers (not possible for an odds ratio). These numbers appear to be mean differences for unit change in hematocrit or mean differences between hematocrit categories (referred to as coefficients in Table 3).
--	--

REVIEWER	Soh, May Ching Oxford University Hospitals NHS Foundation Trust
REVIEW RETURNED	31-May-2021

GENERAL COMMENTS	I would like to congratulate the authors on a well thought out study with methods very clearly described. The results were unsurprising, but the publication of negative findings in research is also important to avoid future repetitions of this study. As a physician, I am not able to comment on the laboratory methods but there are a few pointers that I would like to clarify: 1. The study records pregnancy-induced hypertension at baseline but not pre-eclampsia / eclampsia / HELLP which would likely influence platelet count and even occasionally coagulation parameters. Is there any way this data could be included so that the readers have a better understanding of this? Alternatively, all women with PET/ eclampsia / HELLP should be excluded from the final analysis. 2. The serum ferritin < 15 is somewhat low in pregnancy, and in the units where I have worked, we have used a much higher cut off for oral / parenteral iron supplementation in pregnancy. Is this the accepted ferritin in the institutions where the study was done? 3. There are data on BMI and PPH, with obese women being more prone to PPH. Moreover, obesity increases the risk of iron-deficiency. It would be useful, if possible, to include maternal BMI at the time of the study recruitment as there may be additional factors other than Hb and coagulation markers that contribute to a woman's risk of PPH
--

VERSION 1 – AUTHOR RESPONSE

Reviewer: 1

Dr. K. S. Joseph, Dalhousie Univ

Comments to the Author:

The authors carried out a prospective cohort study in 10 hospitals across India and investigated the association between coagulation indices and anaemia severity among pregnant women, and also between these characteristics and postpartum hemorrhage. 1342 pregnant women were included in the study and there was a 12% loss to follow up. The study showed associations between anaemia severity and D-dimer, fibrinogen, and INR and also postpartum hemorrhage.

Comments

1. This study examined an important global health issue as anaemia in pregnancy is a significant problem and postpartum hemorrhage is common cause of maternal mortality and severe morbidity. Response: We thank the reviewer for highlighting the importance of the study and for the constructive feedback below.

2. In Table 1 and Table 2, it would be preferable to provide the characteristics of the study subjects and follow up information within the categories of interest viz., no/mild anaemia, moderate anaemia and severe anaemia. Such tabular analysis will allow readers to map means and frequencies to the unadjusted mean differences odds ratios from the regression analysis and better understand the results of the adjusted analysis.

Response: We thank the reviewer for the suggestion. Tables 1 and 2 are now revised to present the baseline and follow-characteristics by exposure groups (no/mild anaemia, moderate anaemia and severe anaemia). Median values for the coagulation parameters by anaemia groups were included in Table-3. We have deleted this information from Table-3 to avoid repetition.

3. Of the 1178 subjects followed up and assessed for postpartum hemorrhage, 19 had this outcome. The results of the regression analysis in Table 4 and Table 5 (which involved adjustment for several variables) are limited by this relatively small number of outcomes and this should be acknowledged as a weakness.

Response: We thank the reviewer for the comment. We had acknowledged the reduced statistical power of the analysis for the secondary objective in the limitations section, but have made this clearer in the revised version.

The authors could consider examining blood loss as opposed to postpartum hemorrhage as the outcome.

Response: We thank the reviewer for the thoughtful comment. We considered using blood loss at childbirth as an outcome, but there were two issues –

(i) Women who have Caesarean section normally lose more blood than those who have a normal vaginal birth, so the cut-offs of blood loss for PPH differ for women who had a C-section ($\geq 1000\text{ml}$) and women who had a spontaneous vaginal birth ($\geq 500\text{ml}$). Thus, a woman who lost 800ml of blood during C-section would not be treated as a case of PPH. We could have presented an analysis stratified by mode of birth, to look at the association between quantity of blood loss and Hb in women with C-section versus women with vaginal birth, but the study power did not allow us to undertake this analysis.

(ii) More importantly, in addition to blood loss, we used clinician diagnosed PPH requiring management as another source of information for PPH. This is as per the guidance from ACOG and methods used in other studies recommending the use of more than one source of information for blood loss. As mentioned in the paper, ACOG acknowledges the difficulty in accurately measuring blood loss after childbirth, but recommends use of calibrated drapes and hospital-based protocols for collecting and measuring blood loss after childbirth, which are more accurate than visual estimation¹.

Other comments:

4. Lines 7-9 “The public health problem of anaemia during pregnancy is graded as moderate-severe in 183 countries across the world with about 529 million pregnant women with anaemia in 2011, globally⁵ and an estimated 295,000 maternal deaths annually⁶.”

It is not possible that there were 529 million pregnant women with anaemia in 2011 as currently there are only about 140 million births per year globally. That number refers to women of reproductive age with anaemia. This sentence also needs to be reworded as it implies that the 295,000 maternal deaths may be related to anaemia when in fact that is the total number of maternal deaths per year.
Response: We thank the reviewer for pointing out the error in the numbers. The reviewer has correctly identified that we quoted the total number of women with anaemia in the reproductive age group, rather than total number of pregnant women with anaemia. We have now revised the numbers and have also added the 95% CI from the WHO report. We also agree that the additional information about the annual number of maternal deaths could be confusing and have removed this in the revised draft.

5. A clarification regarding the sample size calculation is warranted. Why did the authors assume equal sized groups with $n_1=n_2$? The primary objective was to compare across categories of anaemia (no/mild, moderate and severe). Also, when the authors refer to ‘change’ (Table S3), do they mean ‘difference’ between groups?

Response: We thank the reviewer for the comments. As mentioned in the introduction section, our primary objective was indeed to investigate the association between blood coagulation parameters and severity of anaemia during the third trimester of pregnancy. Our intention was to have only two groups to examine severity, ‘no/mild’ and ‘moderate/severe’, considering that we did not expect to be able to recruit adequate numbers for a separate severe group within the data collection period, thus the sample size was calculated a priori to measure the difference between two groups with a 1:1 ratio. As mentioned in the ‘sample size’ section of the manuscript, we were able to increase the sample size as we went through the recruitment phase from the original planned 1200 to finally recruiting 1342 pregnant women, which allowed us to examine the difference in the concentration of coagulation parameters between three groups: no/mild anaemia, moderate and severe anaemia. As post-hoc sample size calculations are not considered appropriate, we reported the a priori calculations adding an explanation about the increased sample that allowed us to ultimately examine three instead of two groups of anaemia.

Also, when the authors refer to ‘change’ (Table S3), do they mean ‘difference’ between groups?

Response: Yes, we did mean difference between groups. We have revised the wording in the supplementary table to make this clearer.

6. Column headings in Table S4 state that the numbers provided are odds ratios but several of them are negative numbers (not possible for an odds ratio). These numbers appear to be mean differences for unit change in hematocrit or mean differences between hematocrit categories (referred to as coefficients in Table 3).

Response: We thank the reviewer for pointing out the error. This should have been ‘coefficients’ not ‘odds ratio’. We have corrected this in the revised draft.

Reviewer: 2

Dr. May Ching Soh, Oxford University Hospitals NHS Foundation Trust

Comments to the Author:

I would like to congratulate the authors on a well thought out study with methods very clearly described. The results were unsurprising, but the publication of negative findings in research is also important to avoid future repetitions of this study.

Response: We thank the reviewer for highlighting the importance of the study and for the constructive feedback below.

As a physician, I am not able to comment on the laboratory methods but there are a few pointers that I would like to clarify:

1. The study records pregnancy-induced hypertension at baseline but not pre-eclampsia / eclampsia / HELLP which would likely influence platelet count and even occasionally coagulation parameters. Is there any way this data could be included so that the readers have a better understanding of this? Alternatively, all women with PET/ eclampsia / HELLP should be excluded from the final analysis.

Response: We thank the reviewer for the comment. The variable 'PIH' included all women with gestational hypertension, pre-eclampsia, eclampsia, superimposed pre-eclampsia on chronic hypertension as well as severe forms of pre-eclampsia such as HELLP syndrome. We agree that simply mentioning this group as 'PIH' is confusing so we have now defined this variable clearly in the revised manuscript and re-named it as 'hypertensive disorders of pregnancy'. We chose to adjust for this variable rather than exclude women who have this condition to retain study power. However, we acknowledged possibility of residual confounding by this condition in the discussion section.

2. The serum ferritin < 15 is somewhat low in pregnancy, and in the units where I have worked, we have used a much higher cut off for oral / parenteral iron supplementation in pregnancy. Is this the accepted ferritin in the institutions where the study was done?

Response: We agree with the reviewer that <15 µg/L for ferritin is a conservative cut-off and the WHO guidance is to consider depleted iron stores in pregnancy if ferritin is <30 µg/L if there is high prevalence of iron deficiency and inflammation in the population. The median ferritin level in the study population was 26.6 µg/L so a much higher proportion (52.5%) of women in the study population had ferritin <30 µg/L compared with 28.5% who had ferritin <15 µg/L. Considering the low population median we decided to present the descriptive statistics with the lower cut-off to highlight the high proportion of women who had more severely depleted iron stores.

3. There are data on BMI and PPH, with obese women being more prone to PPH. Moreover, obesity increases the risk of iron-deficiency. It would be useful, if possible, to include maternal BMI at the time of the study recruitment as there may be additional factors other than Hb and coagulation markers that contribute to a woman's risk of PPH

Response: We thank the reviewer for the comment. In the study population, the median BMI at first antenatal visit was 21, with an interquartile range of 18.9 to 23.2 (details added in Table-1 in the revised draft). Thus, a majority of the population were undernourished rather than overweight or obese. Due to the strong correlation between low BMI and anaemia, we did not include maternal BMI in the models. However, we checked the association between BMI and PPH and found that there was no significant association in this study population (OR = 1.04, 95% CI = 0.95 to 1.14, p=0.341), and adding the variable to the multivariable models did not materially change the results.

Reference

1. American College of Obstetricians and Gynecologists. Quantitative Blood Loss in Obstetric Hemorrhage. Washington, DC: ACOG, 2019.